# Role of Renal Biopsy in the Management of Renal Cancer: Concordance between Ultrasound/CT-Guided Biopsy Results and Definitive Pathology, Adverse Events, and Complication Rate

**DOI:** 10.3390/jcm13010031

**Published:** 2023-12-20

**Authors:** Gianmarco Isgrò, Alistair Rogers, Rajan Veeratterapillay, David Rix, Toby Page, Umberto Maestroni, Lorenzo Bertolotti, Francesco Pagnini, Chiara Martini, Massimo De Filippo, Francesco Ziglioli

**Affiliations:** 1Department of Urology, James Cook University Hospital, Middlesbrough TS4 3BW, UK; 2Department of Urology, The Newcastle Upon Tyne Hospitals NHS Foundation Trust, Newcastle upon Tyne NE7 7DN, UKr.veeratterapillay@nhs.net (R.V.);; 3Department of Urology, University Hospital of Parma, 43100 Parma, Italyfziglioli@ao.pr.it (F.Z.); 4Department of Medicine and Surgery, Section of Radiology, University-Hospital of Parma, 43100 Parma, Italymassimo.defilippo@unipr.it (M.D.F.); 5Department of Medicine and Surgery, University-Hospital of Parma, 43100 Parma, Italy

**Keywords:** renal biopsy, kidney cancer, ultrasound-guided biopsy, CT-guided biopsy, small renal masses

## Abstract

(1) Background: In the last decade, the number of detected renal cancer cases has increased, with the highest incidence in Western countries. Although renal biopsy is reported as a safe procedure, it is not adopted in all centres. As it is not possible to accurately distinguish benign tumours using imaging, this may lead to overtreatment. Most of the cancer detected on imaging is treated by surgery, radiofrequency ablation (RFA), or cryotherapy. (2) Methods: This was a single-centre retrospective study of 225 patients studied preoperatively with ultrasound (US)/CT-guided renal biopsy, with the aim of supporting clinical management. Decisions regarding the biopsy were based on either MDT indication or physician preference. US-guided renal biopsy was the first option for all patients; CT-guided biopsy was used when US-guided biopsy was not feasible. The efficacy of renal biopsy in terms of diagnostic performance and the concordance between biopsy results and definitive pathology were investigated. Additionally, adverse events related to the biopsy were recorded and analysed. Data collected throughout the study were analysed using binary logistic regression, Fisher’s exact test, and Pearson’s chi-square test to investigate possible correlations between post-procedural complications and the size of the lesion. (3) Results: Renal biopsy was not diagnostic in 23/225 (10.2%) patients. A CT-guided approach was necessary in 20/225 patients after failure of US-guided biopsy. The complication rate of renal biopsy was 4.8% overall—all Clavien grade I and without any serious sequelae. Interestingly, complications occurred in patients with very different sizes of renal cell carcinoma. No correlation between complications and anticoagulant/antiplatelet drugs was found. No seeding was reported among the patients who underwent partial/radical nephrectomy. (4) Conclusions: Renal biopsy was shown to be safe and effective, with a high concordance between biopsy results and definitive pathology and a low rate of complications. The use of a CT-guided approach whenever the US-guided approach failed improved the diagnostic performance of renal biopsy.

## 1. Introduction

Kidney tumours area major global health concern, with a significant impact on morbidity and mortality rates [1,2].

According to the International Agency for Research on Cancer (IARC), kidney cancer accounted for over 2% of all new cancer cases and 1.8% of cancer-related deaths worldwide in 2020, with a male-to-female ratio of 1.5:1. The incidence of kidney cancer has been increasing over the past few decades, with over 430,000 new cases estimated to be diagnosed globally in 2020. The highest incidence rates are reported in developed countries, with North America and Europe having the highest rates [3].

In the last decade, refinements of imaging techniques have led to an increased number of patients being diagnosed with renal masses [4,5]. However, not all masses are malignant, and some patients may undergo active surveillance only, especially for lesions under 20 mm [6,7].

Percutaneous renal tumour biopsies have become an increasingly popular method for obtaining a histological diagnosis preoperatively, thereby avoiding unnecessary surgery, providing histology prior to ablative treatment, and selecting patients for surveillance appropriately [8,9,10,11].

Therefore, preoperative diagnosis may help patients to avoid not only surgical treatment, but also radiofrequency ablation (RFA) or cryotherapy in selected cases, thus reducing treatment-related morbidity as well as costs for national health systems.

Complications related to renal mass biopsy (RMB) are infrequent and include clinically significant pain (1.2%), gross haematuria (1.0%), pneumothorax (0.6%), and haemorrhage requiring transfusion in the minority of cases (0.4%) [12]. In referral centres, percutaneous core biopsies have low morbidity and are accurate for the diagnosis of malignancy and renal cell carcinoma (RCC) type (LE: 2b). For this reason, preoperative biopsy should be considered whenever possible, as it may influence the management of renal cancer with a low risk of morbidity.

The main issue related to renal biopsy is the risk of not being diagnostic, thus resulting in a waste of time and resources and potentially delaying cancer treatment [13].

We present our series of renal biopsies for renal masses, obtained with an ultrasound (US)/CT-guided approach, which dramatically increased the diagnostic performance of the biopsy. We report the morbidity and adverse events related to the procedure. Finally, we discuss the future potential of the technique in the framework of the multidisciplinary management of renal cancer.

## 2. Materials and Methods

### 2.1. Patients

Freeman Hospital’s Urology Department in Newcastle upon Tyne is a large tertiary referral hospital that serves a population of approximately 2 million individuals. After approval by the local Ethical Authority (Medical Ethical Committee, Newcastel upon Tyne, ref. UR20436/71), between January 2018 and December 2022, a total of 225 consecutive patients underwent renal biopsy for renal tumours. The decision to perform a biopsy was made during a multidisciplinary cancer team meeting or by a single physician, depending on the presentation of the case and the form of referral. The multidisciplinary team discussed the case within 1 week, and the Pathology Department agreed to provide a diagnosis within 10 days to avoid delays in the management of patients.

Patients were referred for biopsy after appropriate imaging (CT scan or MRI) and subsequent classification with a TNM staging system. Data collected included the size (in millimetres), intra-renal location (upper, lower, equatorial, or locally advanced), and side (right, left, or transplanted) of the lesion. After obtaining the biopsy results, the subsequent treatment (partial or radical nephrectomy, thermal ablation, or non-surgical therapy) was based on the type of lesion. In cases of surgical treatment, data on definitive pathology were collected.

### 2.2. Renal Biopsy

US-guided biopsy with a coaxial technique was used as the first approach in most of the patients, whereas CT-guided biopsy was performed in cases of failure of the US-guided technique.

Results and treatment were discussed during the multidisciplinary cancer team meeting and subsequently with the patients, in line with European Association of Urology (EAU) and national guidelines. After the biopsy, patients were observed for 24 h and discharged on day 1 after the biopsy. During the post-operative period, the full blood count was carried out and renal function was checked.

Patients on anticoagulation drugs, antiplatelet drugs, and low-molecular-weight heparin (LMWH) were advised to stop taking their medication to have an appropriate “window” for the procedure. The onset of adverse events was observed following the execution of the renal biopsy. Data on adverse events (none, immediate, or delayed) were collected and classified according to the Clavien–Dindo scale of surgical complications. Finally, data on the diagnostic efficacy of the biopsies were analysed.

### 2.3. Statistical Analysis

Data collected across the study were analysed using binary logistic regression, Fisher’s exact test, and Pearson’s chi-square test to investigate possible correlations between post-procedural complications and the size of the lesion or the concomitant use of anticoagulant or antiplatelet therapy. Additionally, the correlation between tumour size and the diagnostic efficacy of the biopsy was assessed.

## 3. Results

Of 225 renal tumours, 22 were <20 mm, 91 were 20–40 mm, 46 were 40–70 mm, and 49 were >70 mm; in 17 cases, the size was not reported (Table 1).

The total number of US-guided biopsies was 205, while 20 CT-guided biopsies were performed; CT-guided biopsies were employed when US-guided biopsies were not diagnostic (19 cases) or in cases in which preoperative imaging revealed that the puncture was unsafe or at high risk of being unsuccessful (one case). Specifically, one patient was affected by severe polycystic disease, ten had a previous non-diagnostic biopsy, and nine were anatomically unfit for US-guided biopsy.

Overall, 23/225 renal biopsies (10.2%) were “non-diagnostic”. Biopsies were classified as non-diagnostic when the pathology result was “non-diagnostic” (15 cases), “no cancer” (5), “normal” (1), “insufficient sample” (1), or “necrotic tissue only” (1). The reasons for non-diagnostic biopsies are reported in Table 2.

The use of anticoagulant therapy was recorded to explore whether it was correlated with post-biopsy bleeding. We found that 167 patients (74.2%) were not on any form of anticoagulant or antiplatelet drug, while 11 (5.0%) were on anticoagulant drugs, 36 (16%) were on antiplatelet drugs, and 6 (2.7%) were on LMWH; data were missing in 5 cases.

Complications post-renal biopsy was immediate in eight cases (3.5%) and delayed in three (1.3%) (resulting in patients’ re-admission). The complications recorded are reported in Table 3. The overall complication rate was 4.8%. The Clavien complication grade was I for all events (no transfusion was needed in any case). The correlations between complications and lesion size are reported in Table 4.

Only one case of seeding was reported in a cT1a RCC (clear cell), and the patient underwent a robot-assisted partial nephrectomy in 2018. The final pathology showed tumour seeding around the needle trajectory, confirming a T4 stage. The follow-up was negative (29 months).

The T stage distribution revealed 109 pT1a (48.4%), 50 pT1b (22%), 45 pT2 (20%), and 21 pT3/pT4 (9.3%). Of these, 62 patients (32%) underwent surgical treatment, with 41 patients (66.1%) treated with partial nephrectomy and 21 with radical nephrectomy (33.9%). Seven (3.1%) underwent percutaneous RFA.

We found that 156 patients (69%) did not undergo any surgical procedures. Of these, 54 (34.6%) were treated with immunotherapy or chemotherapy, 2 were referred to the Haematology Department (renal lymphoma, 1.2%), 12 (7.7%) received palliative chemotherapy, and 1 received embolization and chemotherapy. In addition, 44 patients (28.2%) are still under surveillance, while 43 patients (27.5%) were lost at follow-up.

Binary logistic regression analysis showed that tumour diameter did not have a significant effect on adverse events, with an odds ratio of 1.007 (95% confidence interval (CI) = 0.993–1.022, *p* = 0.308). Similarly, tumour volume did not have a significant effect on adverse events, with an odds ratio of 1.000 (95% CI = 0.999–1.001; *p* = 0.651). Furthermore, mean tumour diameter did not affect the occurrence of adverse events (*t*-test, *p* = 0.403).

Seven adverse events were registered in patients not on anticoagulants, one in a patient on antiplatelet therapy, one in a patient on LMWH, and none in patients on anticoagulant therapy. The effect of anticoagulant therapy on adverse events was not statistically significant (Fisher’s exact test, *p* = 0.342).

The diameter of the lesion was not correlated within increased risk of complications (Fisher’s exact test showed no significant difference between groups, *p* = 0.069) (Table 4).

The size of the renal mass was correlated with the efficacy of the procedure, with a significantly higher chance of a non-diagnostic biopsy for smaller lesions (Pearson chi-square test, *p* = 0.016) (Table 5).

In the subgroup of 62 patients who underwent surgical treatment (partial or radical nephrectomy), the concordance between the biopsy results and definitive pathology after surgery was analysed. These cases were classified according to lesion location (upper, lower, equatorial, or locally advanced).

Seven patients were excluded because data were not available. Of the remaining 55 patients, 34 (61.8%) showed concordance between the biopsy results and final pathology, while 21 (38.2%) showed discordance. The concordance rates for each location were 60.0% (9/15) for upper, 66.7% (12/18) for lower, 58.8% (10/17) for equatorial, and 60.0% (3/5) for locally advanced lesions.

A chi-square test of independence showed a non-significant relationship between lesion location and concordance (χ^2^ = 2.655, degrees of freedom = 3, *p* = 0.447). The concordance between the biopsy results and definitive pathology are reported in Table 5.

The relationship between tumour size and the concordance between the renal biopsy and definitive pathology results was also investigated. In addition to the seven patients who did not have final pathology data available, there were two patients for whom tumour size data were missing. Of the remaining 53 patients, there was concordance between the histological and final pathology results in 32.

The results of the correlation between renal biopsy efficacy and lesion site and size are reported in Table 6 and Table 7.

Of these, 9.1% (3/33) had a tumour diameter >70 mm, 24.2% (8/33) had a diameter between 40 and 70 mm, and 66.7% (22/33) had a diameter <40 mm. Of the 20 patients with discordance between the histological and final pathology results, 5% (1/20) had a tumour diameter >70 mm, 40% (8/20) had a diameter between 40 and 70mm, and 55% (11/20) had a diameter <40 mm.

No significant correlation between tumour size and concordance was found (χ^2^ = 1.401, *p* = 0.496).

## 4. Discussion

RCC is the most common malignant tumour of the kidney, followed by Wilms’ tumour, which is typically found in children. In the literature, RCC accounts for 85% of renal tumours in adults and can be classified into different subtypes: clear cell (which represents 70–80% of renal tumours), papillary RCC (10–15%), chromophobe carcinoma (5%), and carcinoma of the collecting ducts (<1% of renal epithelial neoplasms). Primary renal tumours arising from the urothelium of the renal pelvis account for approximately 5–10% of cases and range from papillomata to invasive urothelial carcinomas. Among benign tumours, renal papillary adenoma, angiomyolipoma, and oncocytoma are commonly observed.

In the last few years, many authors have reported an increased incidence of renal cancer, which can be explained by the increased detection of small renal masses with radiologic imaging [12,14,15,16]. Indeed, refinements in imaging techniques have led to the early detection of small renal masses, but not all require an active treatment. In this respect, the mass size is reported to be related to the malignancy of the mass: the smaller the mass, the higher the probability that it is less malignant [13,17].

For this reason, small renal masses with uncertain radiologic features on imaging are the most challenging for the clinician, as radical treatment may result in overtreatment, and active surveillance may lead to a delayed diagnosis, anxiety for patients, and an unnecessary cost for the healthcare system [18].

Therefore, it is reasonable to approach uncertain renal masses with all the diagnostic strategies available [19,20,21]. In some cases, the use of both CT scans and MRI may add value to the characterization of the mass; in selected cases, lesion-to-cortex attenuation, and aorta–lesion–attenuation difference (ALAD) on CT were shown to be helpful in differentiating oncocytoma from clear-cell RCC [22]. In recent decades, contrast-enhanced ultrasound (CEUS) of the kidney has also emerged as a cost-effective technique to improve the preoperative diagnosis of renal tumours. In a systematic review and meta-analysis carried out by Tufano et al. [23], this technique was reported to have an accuracy of 0.93 with a negative predictive value of 0.73 in differentiating benign from malignant masses. Similarly, CEUS is reported to be effective at differentiating clear-cell RCC from oncocytoma [24].

Renal biopsy may help make the diagnostic process conclusive, especially when imaging techniques alone fail. Our series is certainly not the first reported on the diagnostic performance of renal biopsies, but to the best of our knowledge, it is one of the largest in the United Kingdom.

The last review of the existing literature on the efficacy of renal biopsy, conducted by Lane et al. in 2008 [21], showed that renal biopsy can provide a definitive diagnosis in 95% of cases. Similarly, in 2013, Menogue et al. [25] not only concluded that renal biopsy is accurate in predicting the malignancy of renal masses, but also maintained that renal biopsy should be considered before surgery whenever the preoperative diagnosis is uncertain.

A meta-analysis comparing RMB with surgical pathology showed that the sensitivity, specificity, and positive predictive value of core RMB are excellent (97%, 94%, and 99%, respectively), making it a reliable method for diagnosing renal malignancies. While histologic characterization of RCC subtypes is highly reliable, the accuracy of renal biopsy for tumour grade varies, with a non-diagnostic rate of 14%, which can be significantly reduced by repeat biopsy. The negative predictive value of RMB is 81%, which suggests that a non-malignant biopsy result may not necessarily indicate the presence of a benign entity [25,26].

Our results confirm that renal biopsy can provide a high level of diagnostic performance if performed properly. Specifically, almost all biopsies were conclusive (202/225), with clear-cell carcinoma and benign lesions being the most common findings, accounting for 38.6% of cases. This high diagnostic accuracy was achieved without any significant adverse events, as the overall complication rate was 4.8%—all Clavien–Dindo grade I and without any serious sequelae.

In the literature, a low incidence of complications is reported. More specifically, authors have reported low rates of gross haematuria (1.0%), pneumothorax (0.6%), and haemorrhage requiring transfusion in the minority of cases (0.4%) [12]. In referral centres, percutaneous core biopsies have low morbidity and are accurate for the diagnosis of malignancy and RCC type (LE: 2b). Our data are in line with the literature, with a very small number of adverse events. Post-operative pain was also considered and was found to be nearly insignificant.

Interestingly, complications occurred in masses of varying sizes, suggesting that there is no correlation between mass volume and the risk of complication.

What sets our cohort apart from previous series is the low rate of failure in achieving a diagnosis. This is likely due to our approach of performing a CT-guided biopsy when a US-guided biopsy was inconclusive.

### 4.1. Safety

#### 4.1.1. Bleeding

Although there are some concerns regarding post-operative bleeding, solid evidence has never been produced. At present, only low-grade adverse events requiring conservative management have been reported [19]. Clinically significant bleeding is unusual and almost always self-limiting, with blood transfusions rarely required. No haematoma had clinical significance in a recent large series [27]. Our series not only confirmed these data, but did so with cases other than large, locally advanced, and metastatic masses.

The tumour at highest risk of bleeding remains the angiomyolipoma. However, the radiologic hallmarks for the diagnosis of this tumour are well known, and in most cases, there is no need for a biopsy to provide a definitive diagnosis. Similarly, the biopsy of cystic tumours may result in collections, ex vacuo bleeding, and, potentially, haematoma.

Regarding the risk of bleeding after biopsy, our data are not conclusive, as statistical analysis on the use of anticoagulant/antiplatelet drugs and their potential association with post-renal biopsy bleeding did not show any statistically significant difference in complications.

#### 4.1.2. Seeding

Although it is quite difficult to draw conclusions on the risk of seeding after renal biopsy due to relatively low-grade evidence, this aspect has been investigated in almost all recent case series. Volpe et al. [27] found that only six cases of seeding from renal parenchymal tumours have been reported. More recently, seven seeding cases were reported in a series of a UK referral centre, and only one of these patients subsequently developed local tumour recurrence at the site of the biopsy [28].

In our series, the seeding rate was far lower, only 0.4%. Therefore, we suggest that the number of reported cases of seeding is affected by the number of patients who are candidates for surgery after renal biopsy, as seeding in patients who undergo surgical excision of RCC is very difficult to document.

One other argument corroborating the assumption that renal biopsy does not carry a clinically significant risk of seeding is that 10/225 biopsies in our series were large transitional cell carcinomas (TCC) that, despite their very different molecular structure and higher seeding abilities [29], did not give rise to local recurrence.

It should be emphasized that renal biopsy does not expose the patient to any adverse oncological event; therefore, the “theoretical” risk should be weighed against the opportunity to avoid definitive treatment (and its potential side effects and morbidity) in patients for whom surveillance is appropriate [30].

### 4.2. Size of Renal Mass

At present, not all centres offer routine renal biopsy to patients presenting with a small renal mass. This is even more true for patients with larger tumours [27].

Interestingly, this paper shows that, for very small masses, renal biopsy might not prove particularly informative, with the non-diagnostic biopsy rate being 8.4% for lesions smaller than 40 mm. This seems to be a limitation, considering that patients with small masses may be the best candidates for surveillance, and in these patients, histology from biopsy would be of utmost importance.

However, our data showed that renal biopsy has a low risk of complication. In addition, the potential of biopsy may be improved by a CT-guided approach whenever a US-guided approach fails, there by greatly increasing its diagnostic performance. In our series, all non-diagnostic lesions with a US-guided biopsy were subsequently re-biopsied using CT-guided approach [26]. We found the diagnostic performance of CT-guided biopsy to be 80% (16/20 biopsies were diagnostic). Not surprisingly, the degree of diagnostic effectiveness achieved when performing biopsies of large renal masses was even higher (about 96% for lesions larger than 40mm).

Conversely, the result of renal biopsy may seem to have allow impact on the management of patients with large masses, as in most cases these patients would be candidates for radical (or partial) nephrectomy if fit for surgery. Nonetheless, in cases in which a large mass is detected on imaging, renal biopsy may be highly informative. Notably, there is scientific evidence suggesting that renal biopsy also plays a role in patients with larger lesions, as well as in locally advanced or even metastatic disease [26,27,31]. In particular, renal biopsy could identify non-clear-cell RCC subtypes in patients with large masses, who may benefit from systemic therapy rather than surgery, and may improve the selection of patients eligible for retroperitoneal lymph node dissection (RPLN) [19,32].

This highlights, once again, the importance of multidisciplinary management to improve the oncologic outcome of patients with renal masses. Therefore, renal biopsy may play a key role in the decision-making process, regardless of the size of the mass.

Finally, the results of biopsies may help to identify those patients with poor outcomes from an oncologic point of view or potentially to engage in shared decision-making with patients about the potentials of different treatment strategies.

## 5. Conclusions

Renal biopsy was proven to be a safe and effective procedure. Adverse events were uncommon. Particularly, bleeding was very infrequent and generally did not require transfusion, and seeding was rare. In our series, renal biopsy showed very satisfactory diagnostic performance. According to the literature, preoperative biopsy should be performed whenever it may influence management of the mass, regardless the size of the mass, due to its low invasiveness. Particularly, in the framework of multidisciplinary management, preoperative biopsy may influence the clinical management of renal masses and potentially help to improve the oncological outcomes of patients diagnosed with renal cancer.

## Figures and Tables

**Table 1 jcm-13-00031-t001:** Patients’ age and lesion diameter (mm) and volume (cm^3^).

	Age (Years)	Lesion Diameter (mm)	Lesion Volume (cm^3^)
Mean	63.6	53.4	274.24
Standard deviation	11.9	40.45	699.69
Median	66	37	26.521
Minimum	18	6	0.11
Maximum	86	210	4849.04
Mean	63.6	53.4	274.24

**Table 2 jcm-13-00031-t002:** Reasons for non-diagnostic biopsies.

15 cases: “non-diagnostic”
5 cases: “no cancer”
1 case: “normal”
1 case: “insufficient sample”
1 case: “necrotic tissue only”

**Table 3 jcm-13-00031-t003:** Lists of adverse events.

Clot retention—haematuria and clot retention post-procedure
2. Haematoma—presented as discomfort post-procedure, US showed fluid, discharged next day, no further complications
3. Urinary retention—required catheter
4. Temperature spike (37.9 °C)
5. Long inpatient stay, issues with dizziness (patient with multiple comorbidities)
6. Immediate perinephric haematoma—patient remained stable, observed overnight
7. Haematuria after 4 days—haemodynamically stable, no clot retention, haemorrhage from liver metastases

**Table 4 jcm-13-00031-t004:** Correlations between adverse events and lesion size.

			No Adverse Event	Adverse Event Occurred	Total
**Coded lesion diameter**	<40 mm	Count	109	4	113
		% within coded lesion diameter	96.5%	3.5%	100.0%
	40–70 mm	Count	50	0	50
		% within coded lesion diameter	100.0%	0.0%	100.0%
	>70 mm	Count	45	4	49
		% within coded lesion diameter	91.8%	8.2%	100.0%
**Total**		Count	204	8	212
		% within coded lesion diameter	96.2%	3.8%	100.0%

**Table 5 jcm-13-00031-t005:** Efficacy of renal biopsy.

			Diagnostic Biopsy	Non-Diagnostic Biopsy	Total
**Coded lesion diameter**	<40 mm	Count	93	18	111
		% within coded lesion diameter	96.0%	16.2%	100.0%
	40–70 mm	Count	48	2	50
		% within coded lesion diameter	96.0%	4.0%	100.0%
	>70 mm	Count	47	2	49
		% within coded lesion diameter	95.9%	4.1%	100.0%
**Total**		Count	188	22	210
		% within coded lesion diameter	89.5%	10.5%	100.0%

**Table 6 jcm-13-00031-t006:** Correlation between renal biopsy efficacy and lesion site.

	Upper	Lower	Equatorial	Locally Advanced	Total
Concordant	9	12	10	3	**3**
Discordant	6	6	7	2	**21**
Missing	2	2	2	1	**7**
**Total**	**17**	**19**	**19**	**6**	**62**

**Table 7 jcm-13-00031-t007:** Correlation between renal biopsy efficacy and lesion size.

	Concordant	Discordant	Total
>70 mm	3	1	4
40–70 mm	8	8	16
<40 mm	22	11	33
**Total**	**33**	**20**	**53**

## Data Availability

Data are available for consultation at the Department of Urology of Freeman Hospital. Please contact Gianmacro Isgrò at gianmarco.isgro1@nhs.net.

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
