# Peer review of "Role of Renal Biopsy in the Management of Renal Cancer: Concordance between Ultrasound/CT-Guided Biopsy Results and Definitive Pathology, Adverse Events, and Complication Rate"

_jcm, 2023, doi:10.3390/jcm13010031_

Round 1

Reviewer 1 Report

Comments and Suggestions for Authors

The article has been significantly improved.  I recommend taking the time to carefully review the text, ensure it is well-formatted.

Author Response

Thank you very much for taking the time to review this manuscript. We are glad you recognized that the manuscript has been highly improved.

This is to inform you that now we made a revision of text, to make it well formatted, as you required.

Thank you for your time.

Reviewer 2 Report

Comments and Suggestions for Authors

I read with great interest the paper on Role of renal biopsy in the management of renal cancer: 

This is certenily an actual debat among urological community,

Authors present their single center experience The sample size is adequate.

Major Suggestions:

- Chi square results should be reported in the tables. Please provide only the p value and not the chi square coefficient.

- Why is Table 1 the last table presented? Moreover, just present the mean (sd) or the median (IQR) basing on the normal or non-normal distribuition of the variable.

- Generally provide more perioperative (pre biopsy) varaiables of the cohort (e.g. can be the RENAL score).

- English revision of the abstract and the introduction

In general also discuss the role of non-invasive preoperative imaging such as CEUS in the discrimination of small renal masses 

doi: 10.3390/diagnostics12102310.

doi: 10.3390/jcm12093070.

Author Response

Response to Reviewer 2

Thank you very much for taking the time to review this manuscript. Please find the detailed responses below. You can also find the corresponding corrections highlighted in the re-submitted file.

Responsse 1.

p values are now more clearly reported.

Response 2.

Table 1 has been moved and is now the first table along the paper. Thank you for pointing this out. Standard deviation is reported, as required in your comment.

About the use of RENAL score or PADUA score, we discussed very much about the appropriateness of reporting it. We eventually decided not to report it because this scores define the complexity of renal tumor when approaching it by surgery (partial nephrectomy). Renal biopsy is a step before surgical treatment of the masss and in the literature we did not found any evidence that che complexity of renal masses for partial nephrectomy correlats with the complexity of renal masses for renal biopsy, let alone that many of the masses biopsied, if positive, were not treated surgically.

We frankly believe that what you pointed out in your comment is of great interest, but still not exactly in the aim of our paper. Considering to design a new study targeted about that would be perfect.

Response 3.

Actually, some discussion about the role of CEUS in the definition of malignant renal masses was lacking. Thank you very much for undelining this weak point of our manuscript. We improved the manuscript accordingly, including this aspect in the discussion. Thank you also for providing literature references to be considered. We also added the references you kindly recommended, which are now cited in the revised version of the manuscript.

Round 2

Reviewer 2 Report

Comments and Suggestions for Authors

The authors have addressed my major concerns.